# Effect of Combustion Chamber Geometrical Parameters on the Decomposition and Combustion Characteristics of an ADN-Based Thruster

**DOI:** 10.3390/mi13040605

**Published:** 2022-04-12

**Authors:** Yangyang Hou, Yusong Yu, Xuhui Liu, Jie Cao

**Affiliations:** 1Hydrogen Energy and Space Propulsion Laboratory, School of Mechanical, Electronic and Control Engineering, Beijing Jiaotong University, Beijing 100044, China; 20116040@bjtu.edu.cn; 2Beijing Institute of Control Engineering, Beijing 100190, China; xhliu99@163.com; 3China North Engine Research Institute, Tianjin 300400, China; jackcao99@163.com

**Keywords:** ammonium dinitramide, chamber geometrical parameters, catalytic combustion, green propellant

## Abstract

In this paper, numerical simulations were used to study the decomposition and combustion processes inside the 0.2 N-class ADN-based thruster, and the effects of two geometrical parameters (length and diameter) of the combustion chamber on the combustion performance were evaluated. The decomposition and combustion processes of the thruster were simulated using a reduced chemical reaction mechanism with 22 components and 20 reactions steps. According to the distribution of the basic physical fields, the variation patterns of the pressure field, velocity field, temperature field, and key component parameters caused by different combustion chamber geometrical parameters were observed and analyzed. The results show that the specific impulse and thrust of the thruster increased and then decreased with the increase of the combustion chamber diameter. When the combustion chamber diameter is 7.9 mm, the specific impulse reaches the maximum value of 206.6 s. Additionally, the specific impulse increased from 186 s to 206 s when the combustion chamber length was changed from 7 mm to 11 mm; the specific impulse increased gradually but not significantly, and the growth trend started to flatten out. The results from the paper can serve as a reference for the design and vacuum testing of an ADN-based thruster.

## 1. Introduction

Hydrazine is widely used in the field of space propulsion, and the high-temperature mixture of ammonia, nitrogen, and hydrogen produced by their catalytic decomposition is ejected through the nozzle to produce a high specific impulse thrust [1]. However, the high vapor pressure and high toxicity of hydrazine lead to high storage, transportation, and operation costs of hydrazine propellants [2]. Therefore, the search for new nontoxic, easily stored, and structurally stable green propellants is a hot research topic in the field of propulsion and power.

The new green single-component liquid propellants mainly include hydroxylamine nitrate (HAN) single-component propellants and ammonium dinitramide (ADN) single-component propellants. ADN-based single-component propellants are high-energy ionic liquids whose high specific impulse and low toxicity are considered to be effective alternatives to hydrazine and among the most promising green propellants in the space propulsion and military fields [3,4]. The solid-state of ADN is unstable and prone to explosions due to its energy-containing substances containing reducing and oxidizing components [5,6,7]. Therefore, dissolving it in water to prepare an ionic solution of a certain concentration for application in green single-component engines can greatly reduce the risk of explosion. To improve the performance of the thruster, researchers added different types of fuels, including methanol and ethanol, by considering the number of oxidant products involved in the catalytic decomposition of liquid ADN-based propellants. Swedish researchers developed an ADN-based propellant with methanol, water, and stabilizer ammonia named LMP-103S [8]. This formulation was successfully used in the High-Performance Green Propellant (HPGP) propulsion system, which was successfully applied to the Prismatic Technology Experiment (PTE) satellite on 15 June 2010, and was demonstrated in a technology demonstration [9].

The combustion of ADN-based liquid propellants is usually characterized by a series of complex physical and chemical processes, and a large number of researchers have investigated the decomposition and combustion processes of the ADN liquid phase. Yu-ichiro Izato et al. [10] proposed a detailed chemical kinetic model of the ADN liquid-phase reaction based on quantum chemical calculations to describe in detail the catalytic decomposition process of ADN-based liquid propellants. Their proposed new model simulates the thermal decomposition of ADN under specific heating conditions and successfully predicts the heat of reaction and the gas produced by decomposition under these conditions. Wingborg et al. [11,12,13,14] tested and evaluated the ignition characteristics, safety, stability, and purity of LMP-103S propellant by studying the ignition mode of ADN-based liquid propellant. Jing et al. [15] conducted an experimental and numerical study of the combustion process in ADN-based thrusters and found that the decomposition of ADN and the oxidation of methanol do not occur simultaneously in the reaction chamber. They pointed out that the decomposition of ADN occurs near the inlet, while methanol is oxidized downstream of the porous medium. Grönland et al. [16] found that the combustion process of aqueous ADN/methanol solution is divided into different reaction stages with different temperatures at the end of each stage. Moreover, they designed and produced catalytic particles with an appropriate activity. Later, Kamal Farhat et al. [17] investigated the catalytic decomposition process of liquid ADN using differential thermal analysis and thermogravimetric analysis methods. They demonstrated that the decomposition process occurs at a lower temperature than the decomposition temperature when no catalyst is used. Rachid et al. [18] also studied the thermal and catalytic decomposition of liquid ADN propellants and found that the addition of catalysts could effectively reduce the initial decomposition temperature of liquid ADN propellants. Meanwhile, Ju Won Kim et al. [19] measured and compared the characteristic properties and performance of different formulations of ADN-based propellants and found that the effect of catalyst active material on the propellant preheating temperature was significantly greater than that of fuel flash point and autoignition temperature.

The catalytic decomposition and combustion processes of ADN-based liquid propellants have been studied by many researchers using numerical simulations. For example, the structure of a catalytic bed filled with catalytic particles has been systematically explored and thoroughly studied [20], as well as the effect of propellant components and ratios on the decomposition and combustion process of thrusters [21]. It can be found that although the abovementioned literature describes the ADN-based propellant decomposition combustion, there are fewer studies on the changes in combustion characteristics of thrusters caused by some specific factors. In the present work, we investigate the effect of combustion chamber geometrical parameters on the spray, evaporation, heat, and mass transfer within the catalytic bed, catalytic decomposition, and combustion processes in a single-component thruster model. The Rosin–Rammler model was used to determine the distribution of ADN droplet particle size. The effective thermal conductivity is added to the energy equation to simulate the complex heat transfer process in porous media. Finally, the decomposition and combustion processes of ADN-based liquid propellants and the effects of different combustion chamber geometrical parameters on the propellant performance are investigated.

The combustion chamber, where the high-temperature combustion reaction of methanol happens, is an important component structure of the ADN-based thruster, and its main geometric parameters play an important role in the internal decomposition and combustion process of the ADN-based thruster. In this paper, numerical simulation is used to carry out the optimized design of ADN-based thruster combustion chamber geometry to provide data reference for the performance optimization of high-performance ADN-based thrusters.

## 2. Description of Numerical Simulations and CFD Methods

### 2.1. Thruster Geometry and Calculation Settings

In this paper, the influence of the combustion chamber geometrical parameters on the internal decomposition and combustion characteristics of the thruster is investigated by three-dimensional numerical simulation. Figure 1a shows the geometric model of the thruster. The DPM method is used to simulate the movement of droplets within the catalytic bed. The resistance of the catalytic bed to the droplet is loaded onto the droplet by the UDF function of the fluent software. The Rosin–Rammler model was used to describe the distribution of droplets within the catalytic bed. The droplet composition is an ADN-based liquid propellant, where the mass fraction ratio of ADN, AN, H_2_O, and CH_3_OH is 50%:14%:18%:18%.

The pressure solver was chosen and was a second-order implicit solver. The apparent velocity solver was used for porous media. The SIMPLE algorithm was used. The outer wall surface of the thruster was a radiation boundary condition. The inlet mass flow rate of the ADN thruster was 0.1 g/s. The porosity and initial temperature of the catalytic bed were 0.275 and 200 °C. The wall temperature was measured by the test, the wall temperature of the catalytic bed was 700 °C, and the wall temperature of the combustion chamber was 900 °C. The geometric parameters of the ADN-based thruster are listed in Table 1.

As shown in Figure 2, the grid independence analysis is carried out using five mesh resolutions: very coarse, coarse, medium, fine, and very fine grid. The average combustion chamber temperature is compared as the basis for the grid independence study. The minimum gird size can be refined down to 2.5 um and 1 um in the nozzle zone for the fine and very fine mesh cases, respectively. The relative derivation is calculated from the results of a very fine grid. The results show that the relative deviations for grid numbers of 20,000, 40,000, 80,000, and 120,000 are 12.3%, 9.25%, 4.61%, and 0.94%, respectively. We think that the relative deviation within 1% has a relatively small effect on the calculation results. When the grid number exceeds 120,000, it can be approximated that the temperature tends to be stable. In this study, we consider a fine grid to be the most appropriate in order to reduce the computational time.

### 2.2. Numerical Description

The structure of the catalytic bed is complex, and the distribution pattern is random. Therefore, the catalytic particles are assumed to be spherical particles in a virtual porous medium in the established model. The droplet distribution model for atomization was selected for numerical simulation from the Rosin–Rammler model. The literature [20] has validated the Rosin–Rammler model as the appropriate spray model for this study. The droplet distribution function of the Rosin–Rammler model is based on the assumption of an exponential relationship between the droplet diameter d and the mass fraction Yd of the droplets with diameters exceeding d. This exponential relationship is as follows.
(1)$$Yd=e^{-{\left(d/\bar{d}\right)}^{n}}$$
where $\bar{d}\;$ is the mean diameter, and n is the propagation parameter used to define the droplet size distribution in the Rosin–Rammler model. The value of n was given in Table 2.

A multicomponent liquid evaporation model was used to describe the evaporation process of ADN-based propellants. In this case, the multicomponent droplet evaporation rate is the result of summing the evaporation rates of individual components [23].

The evaporation rate of component i is calculated by the following equation:(2)$$\frac{dm_{i}}{dt}=A_{p}M_{w,i}k_{c,i}\left(C_{i,s}-C_{i,\infty }\right)$$
where $m_{i}$ is the mass of component i in the droplet, $k_{c,i}$ is the mass transport coefficient of component i, $A_{p}$ is the surface area of the droplet, $M_{w,i}$ is the molecular weight of component i, and $C_{i,s}$ and $C_{i,\infty }\;$are the concentrations of component i on the droplet surface and inside the droplet.

When the total evaporation pressure on the droplet surface exceeds the room pressure, the multicomponent droplet is in a boiling state. The total evaporation pressure is calculated using the formula $P_{t}=\sum P_{i}$, where $P_{i}$ is the partial pressure of component i  [24].

The boiling rate equation:(3)$$\frac{dm_{i}}{dt}=x_{i}\frac{\pi d_{p}k_{\infty }}{c_{p\infty }}\left(2+0.6Re_{d}^{1/2}Pr^{1/3}\right)\ln\left(1+B_{T,i}\right)$$
where $x_{i}$ is the volume fraction of component i in the droplet, $d_{p}$ is the droplet diameter, $k_{\infty }$ is the thermal conductivity of the continuous phase, $c_{p\infty }$ is the specific heat capacity of the continuous phase, and $B_{T,i}=\frac{c_{p\infty }\left(T_{\infty }-T_{p}\right)}{h_{vap,i}}$ is the Spalding heat transfer coefficient of component i.

In order to comprehensively describe the heat exchange process occurring in porous media, the effective thermal conductivity and the corresponding source term are added to the energy equation to model the complex heat transfer process in porous media. Due to the large temperature difference between the solid and fluid phases, the thermal conductivity between the fluid and the solid needs to be considered simultaneously, and a local nonthermal equilibrium model can be used to establish a nonisothermal model for the porous media region [25,26].

Fluid zone energy conservation equation:(4)$$\begin{array}{l} \frac{\partial }{\partial t}(\gamma \rho_{f}E_{f})+\nabla \cdot (\vec{v}\left(\rho_{f}E_{f}+p\right))\\ =\nabla \cdot \left(\gamma k_{f}\nabla T_{f}-(\underset{i}{\sum }h_{i}J_{i})+\left(\vec{\bar{\tau }}\cdot \vec{v}\right)\right)+S_{f}^{h}+h_{fs}A_{fs}\left(T_{f}-T_{s}\right) \end{array}$$

Solid zone energy conservation equation:(5)$$\frac{\partial }{\partial t}((1-\gamma )\rho_{s}E_{s})=\nabla \cdot ((1-\gamma )k_{s}\nabla T_{s})+S_{s}^{h}+h_{fs}A_{fs}(T_{f}-T_{s})$$
where the subscripts f and s represent fluid and solid, γ is the porosity of porous media, $k_{f\;}$is the fluid-phase thermal conductivity, $k_{s}$ is the solid-phase thermal conductivity, $h_{fs}$ is the thermal conductivity between fluid and solid, and $A_{fs}$ is the fluid–solid interfacial density.

It is also necessary to model the heat–fluid–solid coupling in the thruster [27,28].

The solid zone heat transfer conservation equation is:(6)$$\frac{\partial }{\partial t}(\rho h)+\frac{\partial }{\partial x_{i}}(\rho u_{i}h)=\frac{\partial }{\partial x_{i}}\left(k\frac{\partial T}{\partial x_{i}}\right)+q^{m}$$
where ρ is the solid density, h is the sensible enthalpy, k is the solid heat transfer coefficient, T is the temperature, and $q^{m}$ is the volume source term.

Fluid–solid coupling cross-interface equation; similarly, the fluid–solid coupling follows the most basic conservation principle, so at the fluid–solid coupling interface, the fluid–solid heat flow q conservation should be satisfied as follows:(7)$$q_{f}=q_{s}$$

The kinetic model for the catalytic decomposition and combustion reaction of ADN/methanol propellant includes 22 components and 20 primitive reactions (Table 3) [29,30,31]. The equations describing the catalytic reactions are solved using a surface reaction approach. Since the catalytic bed area is modeled using the actual particle filling method, the surface reaction rate can be found by simply solving the energy equation to find the current surface temperature of the particles, giving the activation energy of the surface reaction, the pre-exponential frequency factor, and the surface adsorption enthalpy, and determining the concentration and pressure of the reaction components in the fluid near the particle surface. In order to improve the calculation rate, this study will use the processing of particle surface partitioning to speed up the solution of the surface reaction.

### 2.3. 0.2 N-Class ADN-Based Thruster Vacuum Ground Test

The test data were supported by the Beijing Institute of Control Engineering. Figure 3 shows the vacuum ignition test system of the ADN-based liquid thruster. The thruster is divided into four main parts: solenoid valve, injector, thrust chamber, and thermal control assembly. The solenoid valve realizes the switch control of the propellant; the injector plays the role of blocking the heat of the thrust chamber and carrying out the flow pressure drop control, using a single capillary injection heat-conducting structure. The thrust chamber includes the catalytic bed and the combustion chamber, and the catalytic bed is filled with granular catalyst, which plays the role of catalytic decomposition of the propellant. The noble metal iridium-based catalyst was used as a catalytic bed, and the particle size was 30 mesh. The catalytic particles are loaded into the catalytic bed using a vibratory filling method. The combustion chamber is located downstream of the catalytic bed for further combustion reactions. The thermal control assembly includes an armored heater and a temperature measuring platinum resistor, micro thruster design rated vacuum steady-state thrust 0.2 N, rated flow 0.1 g/s, and rated vacuum-specific impulse greater than 200 s. Green nontoxic ADN propellant consists of ADN, methanol, and water in a certain ratio. When the propellant enters the preheated catalytic bed, ADN decomposes under the catalytic action, and a large amount of oxidizing intermediate products produced by the decomposition of ADN and methanol in the propellant are further burned to release a large amount of heat. The high-temperature gas is ejected through the tail nozzle to generate thrust. The temperature and thrust curves of the thruster are shown in Figure 4 and Figure 5. The case temperatures of the catalytic bed and combustion chamber were measured by the temperature sensors mounted on the thruster at about 720 °C and 1010 °C.

### 2.4. Model Validation

In order to verify the correctness of the thruster model, a comparative study with the temperature of the thruster in the ground vacuum thermal test was conducted. The vacuum ground test data were provided by the Beijing Institute of Control Engineering. The comparison curve of the case temperature of the thruster is shown in Figure 6, and Table 4 shows the comparison results of the numerical simulation and the test under 60 s steady-state conditions. The results show that the combustion chamber temperatures of test and simulation under steady-state conditions are 1013 °C and 1004 °C, with errors less than 5%, which meet the accuracy requirements.

## 3. Results and Discussion

### 3.1. Catalytic Decomposition and Combustion Processes in ADN-Based Thrusters

In this paper, the decomposition reaction of ADN propellant and the oxidizer combustion process in the ADN thruster are simulated. As shown in Figure 7, the change laws of temperature, pressure, velocity, and concentration of various component substances in the thruster are analyzed. As shown in Table 5, the peak temperature of the combustion chamber is 1800 K. The high-temperature region of the combustion chamber is concentrated near the nozzle exit. The region after the heating up, including the back of the catalytic bed and the combustion chamber, is kept in the high-temperature region. In the tapering section of the nozzle, the temperature continues to drop until the gas stream exits the nozzle. The pressure distribution shows that the pressure in the catalytic bed and combustion chamber can reach 0.94 MPa after the complete combustion of ADN/methanol propellant, and the pressure continues to decrease after the nozzle tapering section. The thrust value of the ADN thruster obtained from the simulation is 0.202 N. Compared with the velocity at the nozzle exit, the velocity in the catalytic bed and combustion chamber is lower, and the velocity of the gas passing through the nozzle is accelerated. The maximum velocity at the exit of the nozzle is about 2100 m/s. The velocity increases rapidly at the axial position of the tapering section of the nozzle due to the wall viscosity, and the velocity is lower near the wall of the nozzle. The decomposition reaction of ADN occurs immediately after entering the catalytic bed, and the decomposition rate of ADN near the wall is slower than that in the center of the catalytic bed due to the wall effect, and the ADN is basically consumed in the 1/2~1/3 area of the catalytic bed, and the HN_3_O_4_, methanol, and formic acid produced by the decomposition mainly exist in the front half of the catalytic bed. The reaction products OH were mainly distributed in the combustion chamber and nozzle area, and N_2_O and O_2_ were distributed in the latter half of the catalytic bed and the front part of the combustion chamber.

### 3.2. Effect of Combustion Chamber Diameter on Thruster Decomposition and Combustion Characteristics

The combustion chamber diameter affects the oxidation reaction process inside the combustion chamber and also the propellant flow and heat and mass transfer characteristics inside the combustion chamber. The results of the internal temperature, velocity, and spatial distribution of the main reaction components of the ADN-based thruster at steady-state operation are given in Figure 8. In this work, the decomposition and combustion characteristics of the thruster will be investigated for combustion chamber diameters of 4.9 mm, 5.4 mm, 5.9 mm, 6.4 mm, 6.9 mm, 7.4 mm, 7.9 mm, 8.4 mm, and 8.9 mm.

The results showed that when the combustion chamber diameter was 5.4 mm, the reaction high-temperature zone was distributed near the combustion chamber throat, and the highest peak temperature could reach 1550 K. When the combustion chamber diameter continued to decrease to 4.9 mm, the reaction high-temperature zone spread to the combustion chamber area, and the highest peak temperature was 1450 K. When the diameter of the combustion chamber is 6.4 mm, the high-temperature zone of the reaction is concentrated near the entrance of the combustion chamber, and the highest peak temperature can reach 1600 K. When the diameter of the combustion chamber is 6.9 mm, the temperature distribution is similar to that at 6.4 mm, but the peak temperature reaches 1760 K. When the diameter of the combustion chamber is changed from 7.9 mm to 8.9 mm, the temperature of the combustion chamber starts to decrease and the concentration of carbon oxides in the combustion chamber also decreases. The gas combustion pressure in the combustion chamber is significantly affected by the diameter of the combustion chamber. With the increase in combustion chamber diameter, the combustion pressure gradually decreases [3,32]. The results show that the design of the combustion chamber diameter plays an important role in controlling the temperature distribution. The combustion chamber diameter also has an effect on the component distribution of the ADN-based thruster at different sections. The small diameter of the combustion chamber leads to fast flow, and the methanol in the combustion chamber cannot be completely decomposed [21].

The thruster-specific impulse and thrust values for different combustion chamber diameters are given in Figure 9. The results show that the specific impulse and thrust of the thruster reach the maximum value at the combustion chamber diameter of 7.9 mm. As the combustion chamber diameter increases, the rising trend of the calculated specific impulse and thrust values starts to flatten out. The maximum value of specific impulse was 206.6 s at a combustion chamber diameter of 7.9 mm, and the decrease in combustion chamber diameter has a greater effect on the specific impulse of the thruster than the increase in chamber diameter.

### 3.3. Effects of Combustion Chamber Length on Decomposition and Combustion Characteristics

In order to study the effect of combustion chamber length on the operation of ADN-based thruster, the combustion chamber lengths were set to 5, 7, 9, and 11 mm, respectively. The temperature, velocity, and component distributions of the reactants ADN and intermediate products CO in the thruster axis are given in Figure 10. As the combustion chamber length decreases, the decomposition region of ADN gradually becomes longer, especially when the combustion chamber length is 5 mm, the decomposition region of ADN becomes significantly longer. The change in the combustion chamber length has little effect on the temperature peak of the gas in the combustion chamber, but the area of the high-temperature region increases as the combustion chamber length increases. When the combustion chamber length is 11 mm, at a distance of 0.01 m from the inlet length, the mass of the CO fraction reaches a maximum value of 0.26, the combustion chamber length increases, the gas injection velocity increases, and the gas injection velocity was lowest when the combustion chamber length was 5 mm, while the gas injection velocity was close for the combustion chamber length of 7 mm to 11 mm.

According to Figure 11, it can be seen that when the combustion chamber length is changed from 7 mm to 15 mm, the thrust and specific impulse both increase gradually and then decrease. When the combustion chamber length is 5 mm, the specific impulse and thrust of the thruster decrease significantly, which seriously affects the combustion performance of the thruster.

## 4. Conclusions

In this paper, the effect of combustion chamber geometrical parameters (length, diameter) on the catalyst decomposition and combustion processes in ADN-based thrusters is investigated using numerical simulations. The main conclusions are as follows:(1)The results show that the two geometrical parameters of combustion chamber length and diameter have significant effects on the temperature, pressure, and concentration of key product components in the thruster. By comparing the specific impulse and thrust at various geometrical parameters, it is found that the improvement of thruster performance tends to level off with the increase in combustion chamber length and diameter, and there exists a theoretical optimum design value to achieve the best performance of the thruster;(2)Because of the wall effect, the decomposition rate of an ADN near the wall is slower than that in the center of the catalytic bed, and the decomposition of an ADN is mainly in the catalytic bed area, while the oxidation reaction of methanol is concentrated in the combustion chamber.

## Figures and Tables

**Figure 1 micromachines-13-00605-f001:**
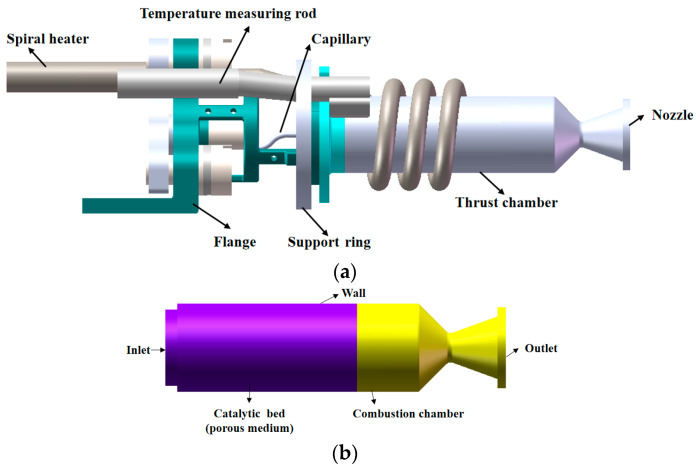
Geometric model of ADN thruster: (**a**) geometric structure; (**b**) computational domain.

**Figure 2 micromachines-13-00605-f002:**
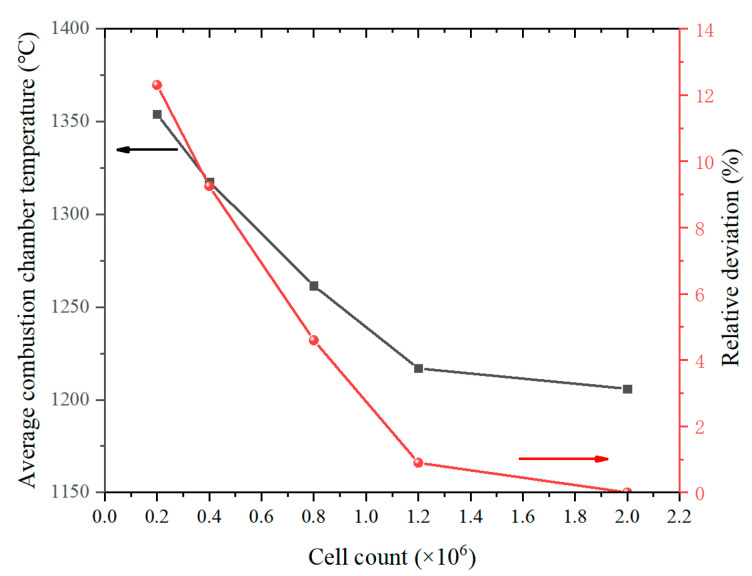
Comparison of average combustion chamber temperature and relative deviation for different cell counts.

**Figure 3 micromachines-13-00605-f003:**
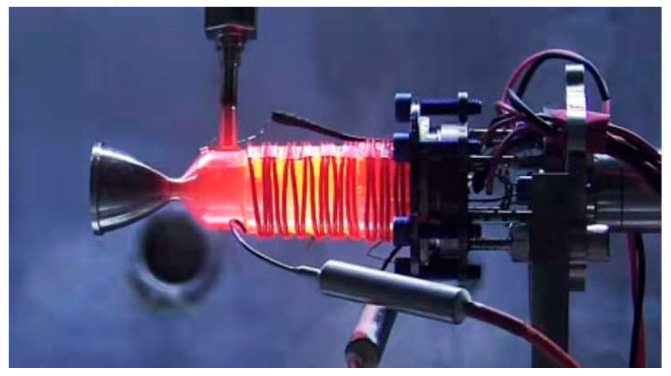
The appearance of the ADN-based thruster including four parts: solenoid valve, injector, catalytic bed, combustion chamber, and nozzle.

**Figure 4 micromachines-13-00605-f004:**
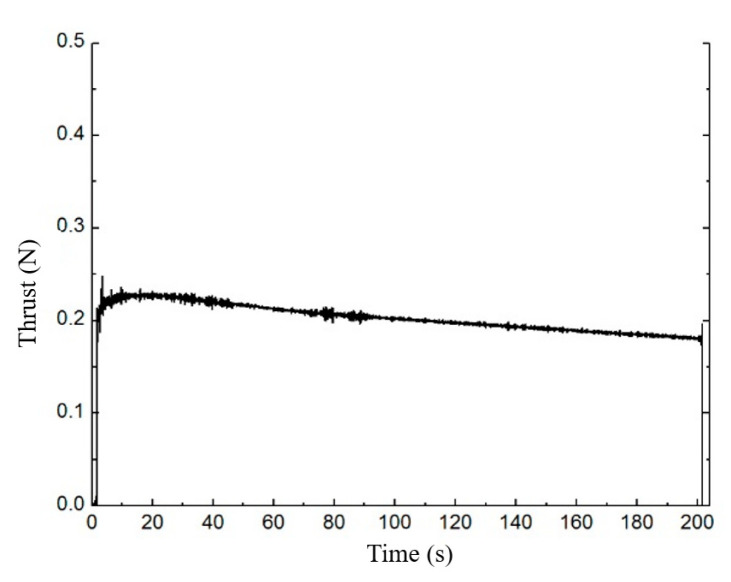
200 s steady-state thrust of 0.2 N-class thruster.

**Figure 5 micromachines-13-00605-f005:**
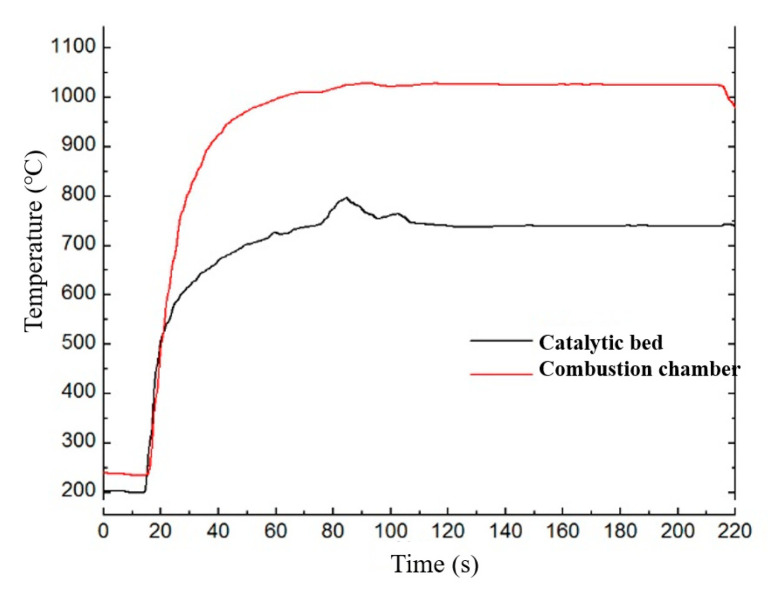
The temperature of catalytic bed and combustion chamber of 0.2 N-class thruster.

**Figure 6 micromachines-13-00605-f006:**
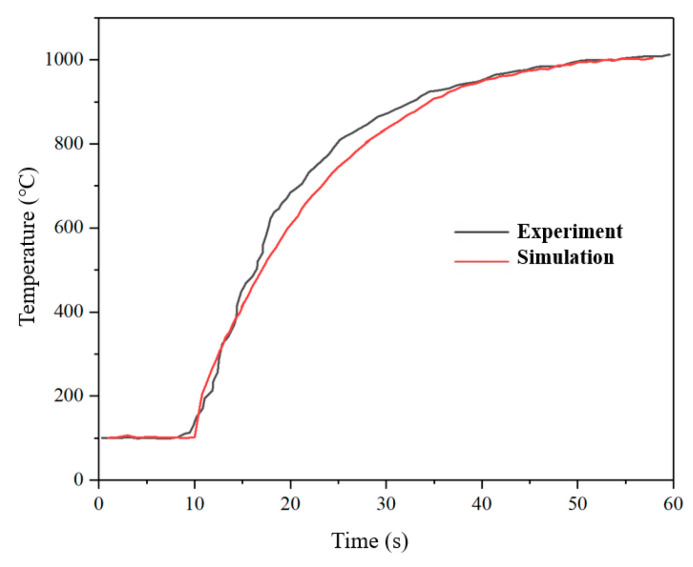
Comparison between simulation and experiment.

**Figure 7 micromachines-13-00605-f007:**
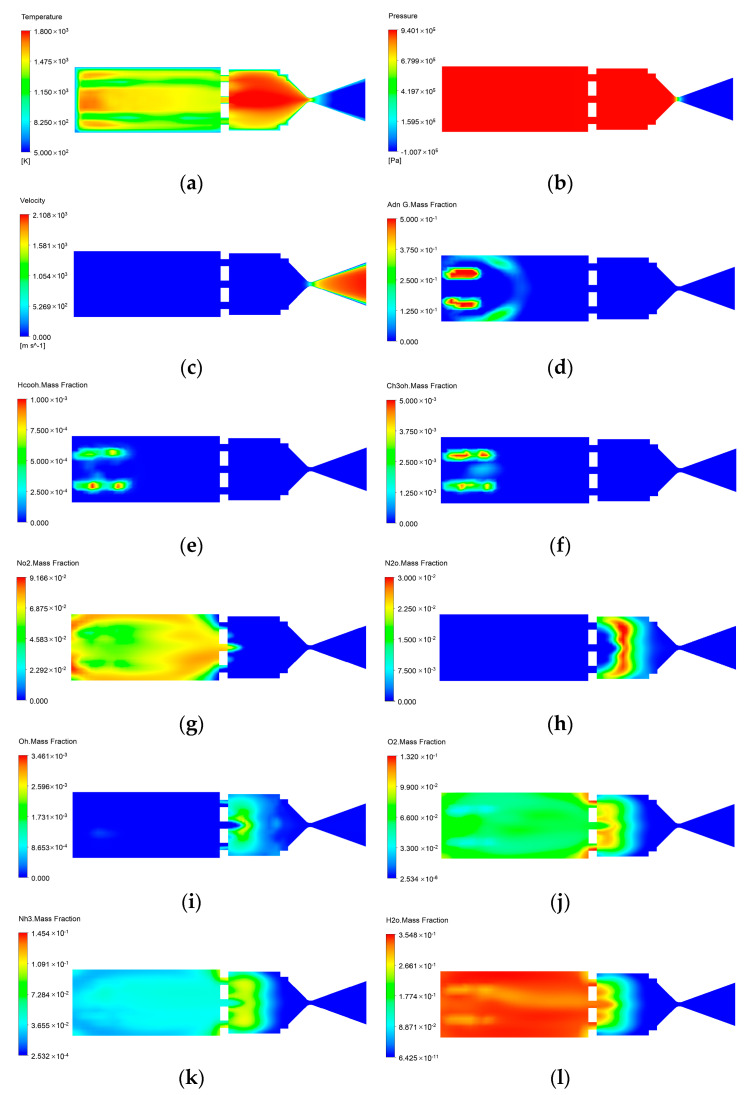
Distributions of important physical fields within the ADN-based thruster: (**a**) temperature; (**b**) pressure; (**c**) velocity; (**d**) ADN mass fraction; (**e**) HCOOH mass fraction; (**f**) CH_3_OH mass fraction; (**g**) NO_2_ mass fraction; (**h**) N_2_O mass fraction; (**i**) OH mass fraction; (**j**) O_2_ mass fraction; (**k**) NH_3_ mass fraction; (**l**) H_2_O mass fraction.

**Figure 8 micromachines-13-00605-f008:**
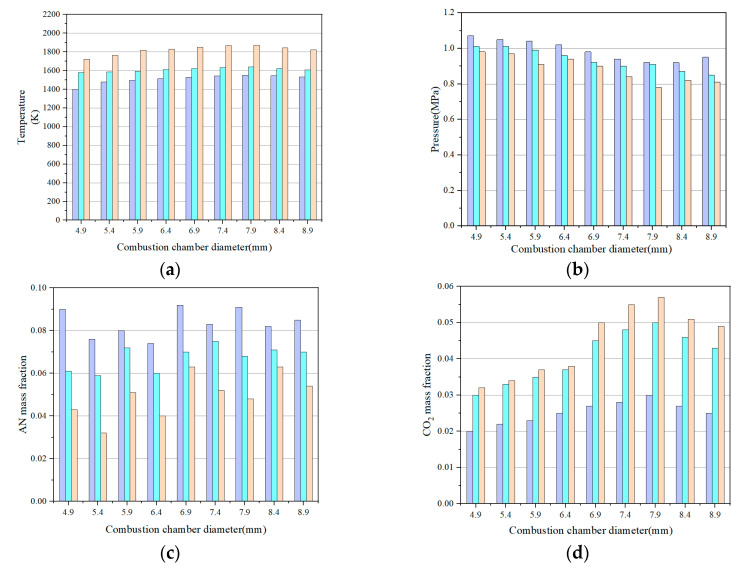
Effect of combustion chamber diameter on temperature, pressure, and concentration of key components: (**a**) temperature; (**b**) pressure; (**c**) AN mass fraction; (**d**) CO_2_ mass fraction; (**e**) O_2_ mass fraction; (**f**) N_2_ mass fraction.

**Figure 9 micromachines-13-00605-f009:**
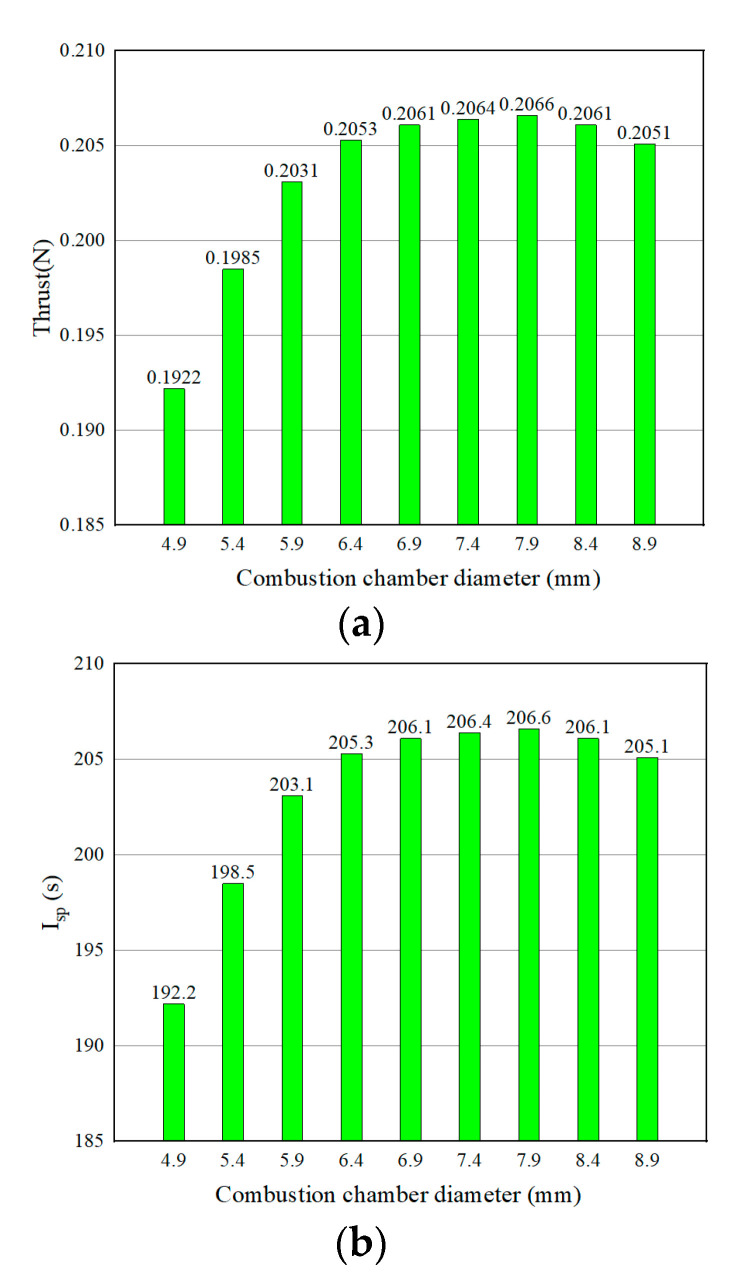
Specific impulse and thrust of thrusters with different combustion chamber diameters: (**a**) specific impulse; (**b**) thrust.

**Figure 10 micromachines-13-00605-f010:**
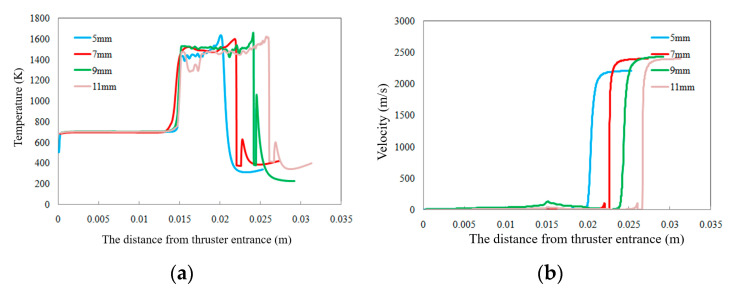
Effect of combustion chamber length on the distribution of components, temperature, and gas velocity along the thruster axis: (**a**) temperature; (**b**) velocity; (**c**) ADN mass fraction; (**d**) CO mass fraction.

**Figure 11 micromachines-13-00605-f011:**
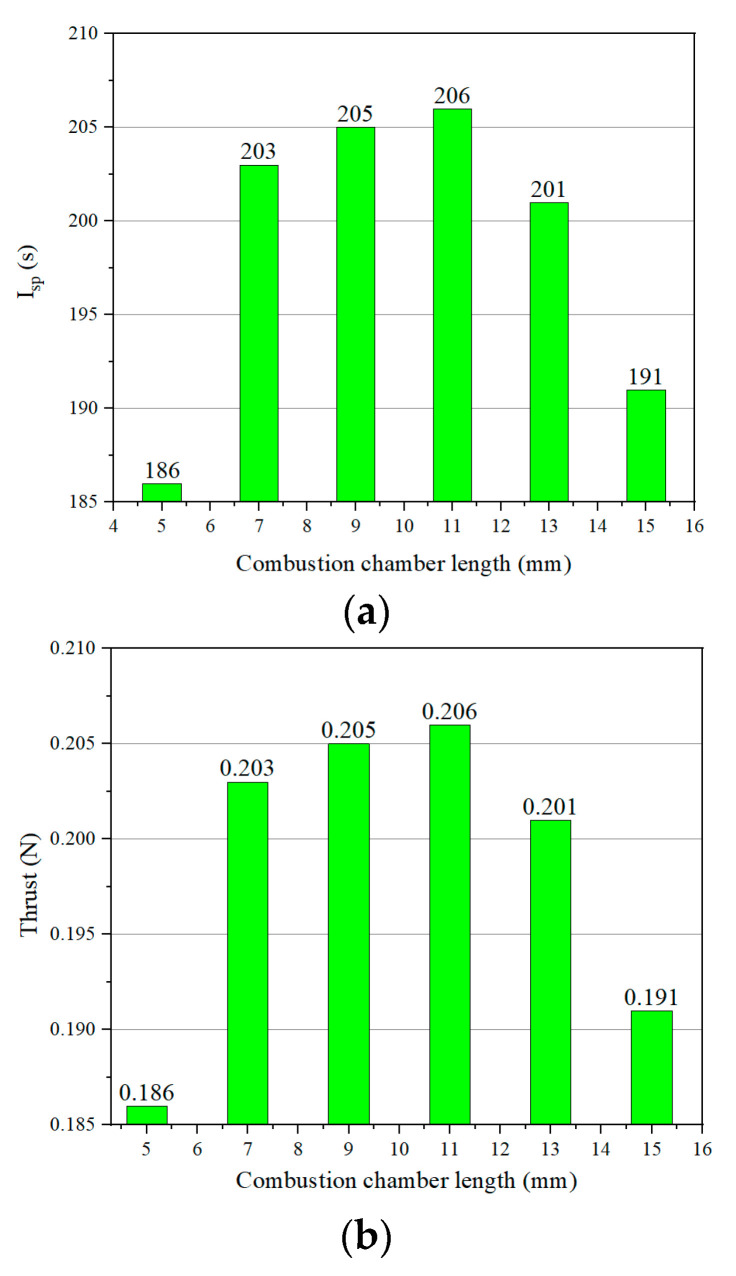
Specific impulse and thrust of different combustion chamber lengths: (**a**) specific impulse; (**b**) thrust.

**Table 1 micromachines-13-00605-t001:** The geometry parameters of ADN-based thruster.

	Parameter	Value
Thruster geometry parameters	Catalyst bed length (m)	1.5 × 10^−2^
	Catalyst bed diameter (m)	6.9 × 10^−3^
	Combustion chamber length (m)	5.3 × 10^−3^
	Combustion chamber diameter (m)	6.9 × 10^−3^
	Nozzle expansion ratio	50

**Table 2 micromachines-13-00605-t002:** The characteristics of the ADN-based propellant and key equation coefficients.

	Value	Literature
Density of liquid (kg/m^3^)	1290	[3]
Cp (Specific Heat) (J/kg·k)	2350	[3]
Viscosity (kg/m·s)	0.0046	[3]
*n*	7.726	[20]
*d* (μm)	80.7	[20]
γ	0.5	[21]
$k_{f}$(W/m·K)	0.8	[22]
$k_{s}$(W/m·K)	25.1	[22]

**Table 3 micromachines-13-00605-t003:** Twenty-step chemical reaction model of ADN/methanol [20,21,30].

ADN(G) + M => NH_3_ + HN_3_O_4_ + M HN_3_O_4_ = HNNO_2_ + NO_2_ 2CH_3_OH + 2NO_2_ = 2HCOOH + N_2_ + 2H_2_O HNNO_2_ + M = N_2_O + OH + M NH_3_ + OH = NH_2_ + H_2_O OH + OH = H_2_O + O HNNO_2_ + OH = H_2_O + 2NO NH_2_ + NO_2_ = H_2_NO + NO NO + NO = N_2_ + O_2_ CH_3_OH + O_2_ = CH_2_OH + HO_2_	CH_3_OH + OH = CH_2_OH + H_2_O CH_2_OH + O = CH_2_O + OH CH_2_O + OH = HCO + H_2_O HCO + OH = H_2_O + CO H_2_NO + O = NH_2_ + O_2_ NH_2_ + NO = N_2_ + H_2_O N_2_O + M = N_2_ + O + M N_2_O + O = N_2_ + O_2_ 2HCOOH + O_2_ = 2CO_2_ + 2H_2_O CO + O + M = CO_2_ + M

**Table 4 micromachines-13-00605-t004:** Comparison between experimental results and simulation results under 60 s steady-state conditions.

	Experimental Result	Numerical Result	Error
Mean thrust (N)	0.21	0.203	3.33%
Temperature of combustion chamber (°C)	1013	1004	0.89%
Temperature of catalytic bed (°C)	706	721	2.12%

**Table 5 micromachines-13-00605-t005:** Key parameters of ADN-based thrusters in the middle of the catalytic bed and the combustion chamber.

	Middle of the Catalytic Bed	Middle of the Combustion Chamber
temperature (℃)	1126	1525
Pressure (MPa)	0.96	0.94
Velocity (m/s)	43	86
ADN mass fraction	0.25	6 × 10^−3^
HCOOH mass fraction	6.1 × 10^−4^	9.4 × 10^−5^
CH_3_OH mass fraction	0.03	0.025
NO_2_ mass fraction	2.4 × 10^−5^	1.3 × 10^−5^
N_2_O mass fraction	3.6 × 10^−3^	5 × 10^−4^
OH mass fraction	5 × 10^−4^	2.3 × 10^−3^
O_2_ mass fraction	4.3 × 10^−5^	1.9 × 10^−5^
NH_3_ mass fraction	0.04	0.073
H_2_O mass fraction	0.28	0.16

## Data Availability

Not applicable.

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
