# Peer review of "Effect of Combustion Chamber Geometrical Parameters on the Decomposition and Combustion Characteristics of an ADN-Based Thruster"

_micromachines, 2022, doi:10.3390/mi13040605_

Round 1

Reviewer 1 Report

The work is devoted to the numerical analysis of how the ADN-fueled thruster geometry affects the propulsion characteristics. The issues raised in the work and the results seem to be relevant however certain revisions should be made before final decision on the presented material.

1) First of all, from my point of view to make novelty and relevance obvious the authors should extend the literature review. Now, there are only two papers from in 2019 and no papers from 2020 and 2021 in the references list. So, the reader can conclude that the issues raised in the Manuscript have lost their relevance.

2) The more detailed description of the problem setup is needed. When reading the Sections 2.1 and 2.2, it is unclear where the solid region is, where the porous medium is, how the droplets are involved in the process, do they move through the porous matrix or they are distributed quiescently inside it, what are the components of each droplet and how do they distributed inside the droplet, what are the wall conditions etc. etc. This part should be revised significantly.

3) The last part of the last paragraph of Introduction is confusing and does not provide explicitly the necessity of the raised issue. The necessity and relevance of the raised problem should be presented clearly!

4) The terminology seems to be incorrect. What is ``bichon''? Since this value is measured in [s], so maybe the authors are speaking about impulse, aren't they? When speaking about ``structural characteristics'' it seems that the authors are speaking about geometrical characteristics while the structure of the thruster remains the same. Authors speak about the ``simplified'' kinetics, isn't it reduced kinetics? What do the authors mean when speaking about ``encrypted'' mesh?

5) Authors cite to the Rosin-Rammler model but do not provide any reference. Moreover, once they write ``rosin-rammler'' but not ``Rosin-Rammler''. Fix please.

6) It is not clear what the numerical instruments do the authors use. Omce they mention ``Fluent'', do they mean that they use Fluent software?

7) It is not clear what is ``Solid heat transfer control'' and what is ``conduction coefficient'' in this equation (eq. 6), is it fluid one or solid one?

8) What do the authors mean when speaking about ``One of the catalytic reactions...''?

9) On figures 3 and 4 the authors introduce three parameters but when showing the validation data only one of them is used on figure 5. Why not to show all three parameters? Herewith, the comparison of thrust seems to be much more important.

10) ``In this thesis'' --> ``In this paper''

11) ``...chamber length is 7 mm to 11 mm'' --> ``...chamber length is changed from 7 mm to 11 mm'' (?)

12) The first sentence in conclusion (1) is not a conlcusion. Please remove it!

13) Conclusion (3) is actually the part of conclusion (1)!

14) The Language should be revised. The grammar is almost ok, but there are some confusing sentences and some repeats in the text.

Author Response

Point 1: First of all, from my point of view to make novelty and relevance obvious the authors should extend the literature review. Now, there are only two papers from in 2019 and no papers from 2020 and 2021 in the references list. So, the reader can conclude that the issues raised in the Manuscript have lost their relevance.

Response 1: Thank you for your comments on this article. It is necessary to ensure the novelty of the article and I have added the latest literature review.

The manuscript is modified as follows:

[7] Prathap kumar Jharapla; G. Vaitheeswaran; M.K. Gupta; R. Mittal. Comparative study of electronic structure, optical prop-erties, lattice dynamics and thermal expansion behaviour of energetic ammonium and potassium dinitramide salts. Mater. Chem. Phys. 2021, 267, 124645.

[19] Ju Won Kim; Seungkwan Back; YeonSoo Jung; Wonjae Yoon; Hong Seop Ban; Sejin Kwon. An alternative ADN based mon-opropellant mixed with tetraglyme. Acta Astronaut. 2021, 178, 241–249.

[26] Qiang Yu; Yihui Yang; Zhifeng Wang; Huibin Zhu. Modeling and parameter sensitivity analysis of fluidized bed solid parti-cle/sCO2 heat exchanger for concentrated solar power plant. Appl. Therm. Engineering 2021, 197, 117429.

Point 2: The more detailed description of the problem setup is needed. When reading the Sections 2.1 and 2.2, it is unclear where the solid region is, where the porous medium is, how the droplets are involved in the process, do they move through the porous matrix or they are distributed quiescently inside it, what are the components of each droplet and how do they distributed inside the droplet, what are the wall conditions etc. etc. This part should be revised significantly.

Response 2: Thank you for your comments on this article. It is a good question. According to your suggestion, the part has been revisd.

The manuscript is modified as follows:

Figure 1 shows the geo-metric model of the thruster. The DPM method is used to simulate the movement of droplets within the catalytic bed. The resistance of the catalytic bed to the droplet is loaded onto the droplet by the UDF function of the fluent software. The Rosin-Rammler model was used to describe the distribution of droplets within the catalyt-ic bed. The droplet composition is ADN-based liquid propellant, where the mass frac-tion ratio of ADN, AN, H2O and CH3OH is 50%:14%:18%:18%.

The pressure solver is chosen and is a second-order implicit solver. The apparent velocity solver is used for porous media. SIMPLE algorithm is used. The outer wall surface of the thruster is a radiation boundary condition. The inlet flow rate of the ADN thruster is 0.1 g/s. The porosity of the catalytic bed is 0.275. The wall temperature is measured by the test, the wall temperature of the catalytic bed is 700℃, and the wall temperature of the combustion chamber is 900℃.

Figure 1. Geometric model of ADN thruster.

Point 3: The last part of the last paragraph of Introduction is confusing and does not provide explicitly the necessity of the raised issue. The necessity and relevance of the raised problem should be presented clearly!

Response 3: Thank you for your comments on this article.

The manuscript is modified as follows:

In the present work, we investigate the effect of combustion chamber structural geo-metrical parameters on the spray, evaporation, heat and mass transfer within the catalytic bed, catalytic decomposition and combustion processes in a single-component thruster model. The influence of porous media on gas-solid surface reactions is considered by introducing specific surface area; numerical simulation methods are used to study the heat transfer characteristics in porous media. The effective thermal conductivity and the corresponding source term are added to the energy equation to simulate the complex heat transfer process in porous media; the heat exchange process occurring in porous media can be described in a more comprehensive way. The Rosin-Rammler model was used to determine the distribution of ADN droplet particle size. The effective thermal conductivity is added to the energy equation to simulate the complex heat transfer process in porous media. Finally, the decomposition and combustion processes of ADN-based liquid propellants and the effects of different combustion chamber geometrical parameters on the propellant performance are investigated.

Point 4: The terminology seems to be incorrect. What is ``bichon''? Since this value is measured in [s], so maybe the authors are speaking about impulse, aren't they? When speaking about ``structural characteristics'' it seems that the authors are speaking about geometrical characteristics while the structure of the thruster remains the same. Authors speak about the ``simplified'' kinetics, isn't it reduced kinetics? What do the authors mean when speaking about ``encrypted'' mesh?

Response 4: Thank you for your comments on this article. I am sorry for the mistakes in words and phrases in the writing of the paper, and I have made the changes as you requested. All errors have been corrected.

Point 5: Authors cite to the Rosin-Rammler model but do not provide any reference. Moreover, once they write ``rosin-rammler'' but not ``Rosin-Rammler''. Fix please.

Response 5: Thank you for your comments on this article. According to your suggestion, I have added references to the Rosin-Rammler model. All errors have been corrected.

The manuscript is modified as follows:

The literature [20] has validated the Rosin-Rammler model as the appropriate spray model for this study.where d is the mean diameter and n is the propagation parameter used to define the droplet size distribution in the rosin-rammler Rosin-Rammler model.

Point 6: It is not clear what the numerical instruments do the authors use. Omce they mention ``Fluent'', do they mean that they use Fluent software?

Response 6: Thank you for your comments on this article. Fluent means Fluent software and I have removed this confusing statement from the paper.

The manuscript is modified as follows:

When the total evaporation pressure on the droplet surface exceeds the room pressure, the multi-component droplet is in a boiling state, and Fluent uses the boiling rate equation by default.

Point 7: It is not clear what is ``Solid heat transfer control'' and what is ``conduction coefficient'' in this equation (eq. 6), is it fluid one or solid one?

Response 7: Thank you for your comments on this article. I’m sorry for errors in writing.

The manuscript is modified as follows:

Solid domain zone heat transfer conservation equation:

 is the solid conduction heat transfer coefficient.

Point 8: What do the authors mean when speaking about ``One of the catalytic reactions...''?

Response 8: Thank you for your comments on this article. I’m sorry for errors in writing.

The manuscript is modified as follows:

One of the The catalytic reactions is solved using a surface reaction approach.

Point 9: On figures 3 and 4 the authors introduce three parameters but when showing the validation data only one of them is used on figure 5. Why not to show all three parameters? Herewith, the comparison of thrust seems to be much more important.

Response 9: Thank you for your comments on this article. It is a good question. According to your suggestion, I have added Table 2.

Table 2. Comparison between experimental results and simulation results under 60s steady-state conditions

Experimental result

Numerical result

Error

Thrust (N)

0.21

0.203

3.33%

Temperature of combustion chamber (°C)

1013

1004

0.89%

Temperature of catalytic bed (°C)

706

721

2.12%

Point 10: ``In this thesis'' --> ``In this paper''

Response 10: Thank you for your comments on this article. All errors were modified.

The manuscript is modified as follows:

In this thesis paper, the decomposition reaction of ADN propellant and the oxidizer combustion process in the ADN thruster are simulated.

Point 11: ``...chamber length is 7 mm to 11 mm'' --> ``...chamber length is changed from 7 mm to 11 mm'' (?)

Response 11: Thank you for your comments on this article. All errors were modified.

The manuscript is modified as follows:

It can be seen that when the combustion chamber length is changed from 7 mm to 11 mm.

Point 12: The first sentence in conclusion (1) is not a conlcusion. Please remove it!

Response 12: Thank you for your comments on this article. According to your suggestion, the first sentence was removed.

Point 13: Conclusion (3) is actually the part of conclusion (1)!

Response 13: Thank you for your comments on this article. According to your suggestion, the conclusion (3) was removed.

Point 14: The Language should be revised. The grammar is almost ok, but there are some confusing sentences and some repeats in the text.

Response 14: Thank you for your comments on this article. I have revised the language of the paper and removed incorrect words and confusing sentences.

Reviewer 2 Report

The subject of manuscript sounds very interesting. Please rewrite the abstract section adding your new findings and exclusive research results. Figure 10 please read it again and find the mistakes in values and description of titles.

Author Response

Point 1: The subject of manuscript sounds very interesting. Please rewrite the abstract section adding your new findings and exclusive research results. Figure 10 please read it again and find the mistakes in values and description of titles.

Response 1: Thank you for your comments on this article. According to your suggestion, the abstract section has been rewritten and new research results have been added. All errors have been corrected.

The manuscript is modified as follows:

Abstract: In this paper, numerical simulations were used to study the decomposition and combustion processes inside the 0.2 N-class ADN-based thruster, and the effects of two geometrical parameters (length and diameter) of the combustion chamber on the combustion performance were evaluated. The decomposition and combustion processes of the thruster were simulated using a reduced chemical reaction mechanism with 22 components and 20 reactions steps. According to the distribution of the basic physical fields, the variation patterns of pressure field, velocity field, temperature field and key component parameters caused by different combustion chamber geometrical parameters were observed and analyzed. The simulation results show that the specific impulse and thrust of the thruster are improved with the increase of the combustion chamber diameter. When increasing the combustion chamber diameter from 4.9 mm to 6.9 mm, the specific impulse increases from 192.2 s to 206.1 s, an increase of 7.2%. And the specific impulse increased from 186 s to 206 s when the combustion chamber length was 7 mm to 11 mm, the specific impulse increased gradually but not significantly and the growth trend starts to flatten out. The results from the paper can serve as a reference for the design and vacuum testing of ADN-based thrusters.

(a)

(b)

Figure 10. Specific impulse and thrust of thrusts with different combustion chamber lengths: (a) specific impulse; (b) thrust.

Reviewer 3 Report

There is a lot of missing information on the simulation condition and variables and the result does not seem to be thoroughly analyzed.

[1] Figure 1 is unclear. The geometry of the thruster should be clarified. For example, what does cross section area A with circles on it mean?

[2] If a heterogeneous catalyst was considered, then detailed catalyst support shape, volume, surface area, etc should be mentioned but catalyst information such as its geometry is missing including how it fill the catalyst bed.

[3] There are several models and equations considered for the simulation. There must be a proper citation for each model and equation.

[4] Each equation has its coefficient. This should be clarified with regard to what value was put for this, and why with a proper citation for each value. There must be a table summarising this.

[5] Catalyst is affected by its reacting temperature and pressure. How was this reflected in the simulation? Where is a catalyst reactivity equation? 

[6] Line 142 "Due to the large temperature difference between the solid and fluid phases, the thermal conductivity between the fluid and the solid needs to be considered simultaneously, and a local non-thermal equilibrium model must be used to establish a non-isothermal model for the porous media region"

How does it affect the relevant equation?

[7] Line 154, "It is also necessary to model the heat-fluid-solid coupling in the thruster." It is missing how this affected the relevant equation.

[8] Grid independency study result is missing.

[9] Regarding Table 1, How were the reaction rate and activation energy of the chemical reactions determined and why?

[10] Figure 2, it shows a heater for the thruster chamber. What was the amount of energy input considered for this for the simulation?

[11] Figure 3, There is varying thrust, which is common for an experimental test because of varying mass flow rate. How was it considered in the simulation?

[12] To make Figure 5, the authors must have found major parameters that affected the simulation accuracy the most. What variables were they in this simulation?

[13] Line 227, "The velocity in the catalytic bed and combustion chamber is low," The expression "low" is unclear. Was it lower than the author expected? Why was it evaluated as low? What was the criteria for that?

[14] Figure 6(a) has a not continuous temperature variation for the upstream part of the catalyst bed. This should be explained properly.

[15] Plane 1, 2, and 3 definitions are missing in figure 6.

[16] Although mass fractions are shown in figure 6, the exact mass fraction should be summarised in a table.

[17] Line 270, "The results show that increasing the combustion chamber diameter is beneficial to the decomposition and combustion of ADN-based propellant."

(a)It is not a logical conclusion without a further study for a much larger diameter. It is normally known that there is an appropriate diameter for each thruster rather than the largest the best.

(b) This must have been affected by the boundary condition of heat flux due to the heater. If the heat flux was constant, then it is so obvious that the larger diameter with a larger chamber surface and larger heat energy input, would increase the thruster performance.

[18] Line 279, "If the combustion chamber diameter is too small, the total heat capacity of the catalytic bed area decreases and.."

Was the total catalyst volume constant? Otherwise, it is not the best simulation reflecting reality. Catalyst volume is a design parameter that determines a decomposable propellant mass flow rate and then aspect ratio should be studied with a fixed catalyst volume, rather than just changing diameter and length without an understanding of catalyst performance.

[19] Line 283, "it also increases the total surface area of the catalytic particles, which enhances the catalytic decomposition reaction and leads ..", Figure 7(b), and Figure 10

Catalyst reactivity is affected by temperature and pressure but this does not seem to be considered in this simulation. Plus, there is an optimum value for the geometrical design of catalyst chamber. For example, if catalyst bed length is extended too much, then there is a pressure drop which degrades thruster performance. But that is not shown in this simulation and this work does not seem to be a proper numerical study with a logical conclusion.

[20] What does "bichon" mean?

[21] Line 299, "The maximum value of bichon was 206.1 s at a chamber diameter of 6.9 mm, and the decrease in chamber diameter had a greater effect on the bichon of the thruster than the increase in chamber diameter."

How was it evaluated compared to a theoretical performance?

[22] Figure 8, It will increase even further with an increasing diameter?

[23] Figure9, How CO mass fraction changed dramatically from around 15% to 0% within around a 5mm range.

Author Response

Point 1: Figure 1 is unclear. The geometry of the thruster should be clarified. For example, what does cross section area A with circles on it mean?

Response 1: Thank you for your comments on this article. Cross-section area A is meant to show the internal mesh of the thruster, but it didn't seem clear, so I decided to delete it.

The manuscript is modified as follows:

Figure 1. Geometric model of ADN thruster.

Point 2: If a heterogeneous catalyst was considered, then detailed catalyst support shape, volume, surface area, etc should be mentioned but catalyst information such as its geometry is missing including how it fill the catalyst bed.

Response 2: Thank you for your comments on this article. The noble metal iridium-based catalyst was used as catalytic bed and the particle size is 30 mesh. The catalytic particles are loaded into the catalytic bed using a vibratory filling method. The surface-to-volume ratio of the catalyst particles is about 2500. The bulk reaction is currently used to approximate the complex surface and volume reactions in the actual catalytic bed, so that no specific carrier shape is required in the simulation. The key parameter for the simulation is the porosity, which affects the fluid-solid heat transfer within the catalytic bed. The porosity of the catalytic bed is 0.275. Pictures of the catalytic particles are shown below:

Physical picture of catalyst particles

SEM of catalyst particles

Point 3: There are several models and equations considered for the simulation. There must be a proper citation for each model and equation.

Response 3: Thank you for your comments on this article. The models and equations have been added to the references.

The manuscript is modified as follows:

The literature [20] has validated the Rosin-Rammler model as the appropriate spray model for this study.

In this case, the multi-component droplet evaporation rate is the result of summing the evaporation rates of individual components [22].

The total evaporation pressure is calculated using the formula , where  is the partial pressure of component [23].

A local non-thermal equilibrium model can be used to establish a non-isothermal model for the porous media region [24,25].

It is also necessary to model the heat-fluid-solid coupling in the thruster [26]

Point 4: Each equation has its coefficient. This should be clarified with regard to what value was put for this, and why with a proper citation for each value. There must be a table summarising this.

Response 4: Thank you for your comments on this article. It is a good question. According to your suggestion, I have added a new table.

The manuscript is modified as follows:

Nomenclature

AN

hydroxylamine nitrate

concentrations of component  on the droplet surface

mass fraction of the droplets

concentrations of component  inside the droplet

mean diameter

pressure of evaporation

propagation parameter

volume fraction of component  in the droplet

the mass of component  in the droplet

droplet diameter

mass transport coefficient of component

thermal conductivity of the continuous phase

surface area of the droplet

specific heat capacity of the continuous phase

molecular weight of component

fluid

solid

porosity of porous media

fluid phase thermal conductivity

solid phase thermal conductivity

thermal conductivity between fluid and solid

fluid-solid interfacial density

solid density

sensible enthalpy

solid heat transfer coefficient

solid temperature

volume source term

heat flow

heat flow of fluid

heat flow of solid

specific impulse

DPM

discrete phase model

UDF

User defined function

Point 5: Catalyst is affected by its reacting temperature and pressure. How was this reflected in the simulation? Where is a catalyst reactivity equation?

Response 5: Thank you for your comments on this article. In this paper, we neglected the catalyst reactivity equation. Volumetric reactions were used to approximate the catalytic and volumetric chemical reactions within the catalytic bed. In the next study, we will take your comments and consider the effect of the catalyst reactivity equation.

Point 6: Line 142 "Due to the large temperature difference between the solid and fluid phases, the thermal conductivity between the fluid and the solid needs to be considered simultaneously, and a local non-thermal equilibrium model must be used to establish a non-isothermal model for the porous media region" How does it affect the relevant equation?

Response 6: Thank you for your comments on this article. The local none thermal equilibrium model is equation 4 and equation 5, whose main purpose is to find out the temperature distribution of the fluid. The advantage of the local non-heat balance model compared with the local heat balance equation is that the temperature distribution of the fluid and the solid can be distinguished and the calculation is more realistic.

Point 7: Line 154, "It is also necessary to model the heat-fluid-solid coupling in the thruster." It is missing how this affected the relevant equation.

Response 7: Thank you for your comments on this article. The meaning of this sentence is to solve for the non-catalytic bed area, and finally obtain the heat flow and temperature distribution of the combustion chamber and nozzle.

Point 8: Grid independency study result is missing.

Response 8: Thank you for your comments on this article. I have added the grid independency study result.

The manuscript is modified as follows:

As shown in Figure 2, the grid independence analysis is carried out using five mesh resolutions: very coarse, coarse, medium, fine and very fine grid. The minimum gird size can be refined down to 2.5 um and 1 um in the nozzle zone for the fine and very fine mesh cases, respectively. Relative derivation, calculated from the results of a very fine grid. In this study, we consider a medium grid to be the most appropriate in order to reduce the computational time.

Figure 2. Comparison of combustion chamber temperature and relative deviation for different cell counts.

Point 9: Regarding Table 1, How were the reaction rate and activation energy of the chemical reactions determined and why?

Response 9: Thank you for your comments on this article. The finite rate chemical reaction model

with 20 reactions and 22 species has been employed based upon the valuable experiences contributed by the literatures [27–29].

[27] Gunawan R; Zhang D. Thermal stability and kinetics of decomposition of ammonium nitrate in the presence of pyrite. J. Hazard. Mater. 2009, 165, 751–758.

[28] Korobeinichev OP; Bolshova TA; Paletsky AA. Modeling the chemical reactions of ammonium dinitramide (ADN) in a flame. Combust. Flame 2001, 126, 1516–1523.

[29] Foustoukos DI; Stern JC. Oxidation pathways for formic acid under low temperature hydrothermal conditions: implications for the chemical and isotopic evolution of organics on Mars. Geochim. Cosmochim. Acta 2012, 76, 14–28.

Point 10: Figure 2, it shows a heater for the thruster chamber. What was the amount of energy input considered for this for the simulation?

Response 10: Thank you for your comments on this article. In experiments, the catalytic bed usually needs to be preheated to 200°C with an electric heating wire. And in the simulation, we achieve the initialization temperature by using the patch function of fluent software.

Point 11: Figure 3, There is varying thrust, which is common for an experimental test because of varying mass flow rate. How was it considered in the simulation?

Response 11: Thank you for your comments on this article. It is a good question. Due to the complex chemical and physical processes in the ADN- based thruster, we use the average mass flow rate as the boundary condition for the simulation to simplify the calculation.

Point 12: To make Figure 5, the authors must have found major parameters that affected the simulation accuracy the most. What variables were they in this simulation?

Response 12: Thank you for your comments on this article. Since the combustion performance of the thruster is mainly determined by the chemical reaction of the ADN-based propellant, the temperature can be used as the main parameter to validate the model. In order to compare the accuracy of the models comprehensively, I have added Table 2.

Table 2. Comparison between experimental results and simulation results under 60s steady-state conditions

Experimental result

Numerical result

Error

Thrust (N)

0.21

0.203

3.33%

Temperature of combustion chamber (°C)

1013

1004

0.89%

Temperature of catalytic bed (°C)

706

721

2.12%

Point 13: Line 227, "The velocity in the catalytic bed and combustion chamber is low," The expression "low" is unclear. Was it lower than the author expected? Why was it evaluated as low? What was the criteria for that?

Response 13: Thank you for your comments on this article. Here is an error caused by the inaccuracy of the authors' description. The velocity in the catalytic bed and combustion chamber is much lower than the velocity at the nozzle exit, which is 2100 m/s at the nozzle, while the maximum velocity in the catalytic bed and combustion chamber is only 100 m/s.

The manuscript is modified as follows:

The velocity in the catalytic bed and combustion chamber is low, and the gas velocity is accelerated through the nozzle. Compared to the velocity at the nozzle exit, the ve-locity in the catalytic bed and combustion chamber is lower and the velocity of the gas passing through the nozzle is accelerated.

Point 14: Figure 6(a) has a not continuous temperature variation for the upstream part of the catalyst bed. This should be explained properly.

Response 14: Thank you for your comments on this article. The thruster inlet is set up with four nozzles, and we guess that the temperature drops due to the evaporation of the propellant spray which absorbs heat. And the temperature in other areas of the catalytic bed increases because a catalytic decomposition reaction occurs. Therefore, there is a discontinuous temperature change.

Point 15: Plane 1, 2, and 3 definitions are missing in figure 6.

Response 15: Thank you for your comments on this article. Planes 1, 2, and 3 refer to the catalytic bed , the combustion chamber  and the nozzle , respectively. Here the author refers to the transverse section of the ADN-based thruster. This is a mistake caused by improper operation during the drawing, which has been corrected now.

The manuscript is modified as follows:

(a)

(b)

(c)

(d)

(e)

(f)

(g)

(h)

(i)

(j)

(k)

(l)

Point 16: Although mass fractions are shown in figure 6, the exact mass fraction should be summarised in a table.

Response 16: Thank you for your comments on this article. The table 3 has been added.

Table 3. Key parameters of ADN-based thrusters in the middle of the catalytic bed and the combustion chamber

Middle of the catalytic bed

Middle of the combustion chamber

temperature (℃)

1126

1525

Pressure (MPa)

0.96

0.94

Velocity (m/s)

43

86

ADN mass fraction

0.25

6×10-3

HCOOH mass fraction

6.1×10-4

9.4×10-5

CH3OH mass fraction

0.03

0.025

NO2 mass fraction

2.4×10-5

1.3×10-5

N2O mass fraction

3.6×10-3

5×10-4

OH mass fraction

5×10-4

2.3×10-3

O2 mass fraction

4.3×10-5

1.9×10-5

NH3 mass fraction

0.04

0.073

H2O mass fraction

0.28

0.16

Point 17: Line 270, "The results show that increasing the combustion chamber diameter is beneficial to the decomposition and combustion of ADN-based propellant."

(a)It is not a logical conclusion without a further study for a much larger diameter. It is normally known that there is an appropriate diameter for each thruster rather than the largest the best.

(b) This must have been affected by the boundary condition of heat flux due to the heater. If the heat flux was constant, then it is so obvious that the larger diameter with a larger chamber surface and larger heat energy input, would increase the thruster performance.

Response 17: Thank you for your comments on this article. Obviously your comment is correct. We did additional cases for much larger diameters (7.4mm and 7.9mm), and found that the specific impulse and thrust increased with increasing diameter, but the growth trend starts to flatten out. We speculate that there is a theoretical diameter suitable for the thruster, but we do not have specific results at this time. In further studies, we will discuss the effect of the aspect ratio on the combustion performance in conjunction with experimental data.

Point 18: Line 279, "If the combustion chamber diameter is too small, the total heat capacity of the catalytic bed area decreases and.."

Was the total catalyst volume constant? Otherwise, it is not the best simulation reflecting reality. Catalyst volume is a design parameter that determines a decomposable propellant mass flow rate and then aspect ratio should be studied with a fixed catalyst volume, rather than just changing diameter and length without an understanding of catalyst performance.

Response 18: Thank you for your comments on this article. Your comments is very reasonable. The total volume of the catalyst should be fixed when the mass flow rate is constant. And changing the diameter of the combustion chamber does not affect the total heat capacity of the catalytic bed area. In this paper, we simply discuss the effect of combustion chamber diameter on combustion performance, and keeping the catalytic bed diameter constant. All errors have been corrected.

The manuscript is modified as follows:

With the increase of combustion chamber diameter, the combustion pressure gradually increases, and the combustion pressure decreases significantly when the combustion chamber diameter is reduced. The results show that increasing the combustion cham-ber diameter is beneficial to the decomposition and combustion of ADN-based propel-lant. It can be seen that the design of the combustion chamber diameter plays an im-portant role in controlling the temperature distribution. Reducing the combustion chamber diameter increases the average flow rate of the propellant in the catalytic bed combustion chamber and reduces the stagnation time scale. Increasing the diameter of the combustion chamber means a larger heat input, which enhances the catalytic de-composition reaction and leads to an excessively rapid decomposition of ADN exothermally and accelerates the combustion reaction with CH3OH, releasing a large amount of heat.

Point 19: Line 283, "it also increases the total surface area of the catalytic particles, which enhances the catalytic decomposition reaction and leads ..", Figure 7(b), and Figure 10

Catalyst reactivity is affected by temperature and pressure but this does not seem to be considered in this simulation. Plus, there is an optimum value for the geometrical design of catalyst chamber. For example, if catalyst bed length is extended too much, then there is a pressure drop which degrades thruster performance. But that is not shown in this simulation and this work does not seem to be a proper numerical study with a logical conclusion.

Response 19: Thank you for your comments on this article. Catalyst reactivity is affected by temperature and pressure. In the experiment, we found that for the factors affecting the temperature of the thruster is mainly the exothermic process of combustion in the combustion chamber, and increasing the length and diameter of the combustion chamber will enhance the specific impulse to a certain extent. However, due to the particularly strict structural requirements of the thruster in the vacuum test, the experiment was subject to development. In the simulation, we simply study the effect of geometric parameters of the combustion chamber on the combustion performance of the thruster, not the geometric parameters of the catalytic bed chamber. This is a misunderstanding caused by the author's writing error. In the next study, we will no longer consider the effect of a single factor on the impulse and thrust, and your proposed aspect ratio has given us great guidance.

Point 20: What does "bichon" mean?

Response 20: Thank you for your comments on this article. "bichon" should be modified to "Specific impluse". I am sorry for the mistakes in words and phrases in the writing of the paper, and all errors have been corrected.

Point 21: Line 299, "The maximum value of bichon was 206.1 s at a chamber diameter of 6.9 mm, and the decrease in chamber diameter had a greater effect on the bichon of the thruster than the increase in chamber diameter."

How was it evaluated compared to a theoretical performance?

Response 21: Thank you for your comments on this article. The simulation results show that increasing the diameter of the combustion chamber is effective in increasing the specific impulse, but the growth trend starts to flatten out. For example, if the diameter of the combustion chamber is 6.9 mm, the specific impulse increases by 1.1% when the diameter of the chamber is increased by 5 mm. And when the chamber diameter is decreased by 5 mm, the specific impulse is decreased by 2.3%.

Point 22: Figure 8, It will increase even further with an increasing diameter?

Response 22: Thank you for your comments on this article. We did additional cases for diameters of 7.4mm and 7.9mm, and found that the specific impulse and thrust increased with increasing diameter, but the increase was particularly small, less than 1%. We speculate that increasing the diameter of the combustion chamber will not effectively increase the diameter of the combustion chamber.

Point 23: Figure 9, How CO mass fraction changed dramatically from around 15% to 0% within around a 5 mm range.

Response 23: Thank you for your comments on this article. Due to the complex chemical and physical processes in the ADN- based thruster, We speculate that this is due to the constant generation and consumption of CO. The reaction equation for CO formation:HCO+OH=H2O+CO, Reaction equation for CO consumption:CO+O+M=CO2+M.

ADN(G)+M=>NH3+HN3O4+M

HN3O4=HNNO2+NO2

2CH3OH+2NO2=2HCOOH+N2+2H2O

HNNO2+M=N2O+OH+M

NH3+OH=NH2+H2O

OH+OH=H2O+O

HNNO2+OH=H2O+2NO

NH2+NO2=H2NO+NO

NO+NO=N2+O2

CH3OH+O2=CH2OH+HO2

CH3OH+OH=CH2OH+H2O

CH2OH+O=CH2O+OH

CH2O+OH=HCO+H2O

HCO+OH=H2O+CO

H2NO+O=NH2+O2

NH2+NO=N2+H2O

N2O+M=N2+O+M

N2O+O=N2+O2

2HCOOH+O2=2CO2+2H2O

CO+O+M=CO2+M

Round 2

Reviewer 1 Report

After the revision the Manuscript became much more readable. I think that the work deserves to be published after minor Language chekings and revisions according to the following comments:

1) I think that the authors did not catch what I wanted to say in my comment 3. I meant that they did not write clearly what are the necessity and relevance of the problem that they were solving. So, for the reader it is not clear is this problem setup has sence or not. And in their revised version they are writing about the model, about the calculations, but what is the particular purpose of this work?

2) "The catalytic reactions is solved" --> "The equations describing the catalytic reactions are solved"

Author Response

Thank you for your comments on this article. Please see the attachment for the specific response comments

Reviewer 3 Report

[1] Previous comment #1 "Figure 1 is unclear. The geometry of the thruster should be clarified. For example, what does cross section area A with circles on it mean?" has not been answered properly. The detailed geometry of the thruster is missing, such as the circles on the area A previously, what's inside of the thruster, dimensions etc.

[2] Previous comment #4 "Each equation has its coefficient. This should be clarified with regard to what value was put for this, and why with a proper citation for each value. There must be a table summarising this." also has not been answered properly. What this reviewer is concerned about is potential confusion from readers of this article on the value of coefficients that can be in fact also arbitrarily chosen to be used in this simulation. There is always a coefficient that makes the simulation result close to experimental data. In the simulation section, there should be a table that shows what values were practically used as inputs for the parameters/coefficients and why with proper citations, or in case there were any inputs that were determined by the authors based on the experimental data as a reference, the authors should clarify it.

[3] Regarding 'Grid independency study', the figure added shows the temperature that is still varying. The authors should add discussions on how the result can be determined as grid independency has been acquired.

[4] Regarding previous comment #10, "Figure 2, it shows a heater for the thruster chamber. What was the amount of energy input considered for this for the simulation?" that has not been answered properly, Was the constant temperature considered as the boundary condition? or initial temperature and constant heat flux?

[5] Regarding previous comment #11 "Figure 3, There is varying thrust, which is common for an experimental test because of varying mass flow rate. How was it considered in the simulation?" If an averaged constant mass flow rate was considered, what was it? and how in that case thrust is only 3.33% different from the experimental result for simulation? 

[6] Regarding previous comment #14 "Figure 6(a) has a not continuous temperature variation for the upstream part of the catalyst bed. This should be explained properly." if uniform catalyst structures and fillings were considered for the catalyst bed, isn't it supposed to have uniform temperature distribution? But as shown in the figure, there are some non-uniform temperature distributions and the physical results should be analysed properly.

[7] Regarding "Table 3. Key parameters of ADN-based thrusters in the middle of the catalytic bed and the combustion" Is the summation of all the elements the same as the unit flow or total mass flow rate? Also , is it not supposed to be the same between the catalyst bed and combustion chamber?

[8] Regarding the authors' reply "Response 18: ..The manuscript is modified as follows:

With the increase of combustion chamber diameter, the combustion pressure gradually increases, and the combustion pressure decreases significantly when the combustion chamber diameter is reduced. The results show that increasing the combustion cham-ber diameter is beneficial to the decomposition and combustion of ADN-based propel-lant. It can be seen that the design of the combustion chamber diameter plays an im-portant role in controlling the temperature distribution.

Reducing the combustion chamber diameter increases the average flow rate of the propellant in the catalytic bed combustion chamber and reduces the stagnation time scale.

Increasing the diameter of the combustion chamber means a larger heat input, which enhances the catalytic de-composition reaction and leads to an excessively rapid decomposition of ADN exothermally and accelerates the combustion reaction with CH3OH, releasing a large amount of heat."

These three parts seem to be against physics. The authors should find relevant references and compare their results, to make the analysis more physically correct and logical.

Author Response

(The authors gave the same response as above.)

Round 3

Reviewer 3 Report

  1. Still there is unclarity on what was simulated with regard to its geometry. Potential readers of this article would be confused finding the geometry of figure 1 is different from simulated results in figure 4.
  2. Regarding the grid independent study result (combustion chamber result with respect to the number of cell), the newly provided one look different from the previous one. Data that is varying without any support, would seem to have low reliability.
  3. The boundary condition of this simulation is unclear. The initial constant temperature that the author mentioned should be clarified with regard to its geometry and from where to where on the thruster the heater was assumed to be located.
  4. The unclarity issue of geometry also can be found in the previous authors' answer with an image for the injector, hard to understand what is what without any labels on it and it is even not mentioned in the article. The geometry in figure 1 should be modified including the injector component as well for better clarity.
  5. Regarding previous comment #7 on table 3. Even if there are figures only for interested molecules that are acceptable, a table is believed to contain all the elements. Otherwise, how could the author have validated the simulation result without checking the summation of all the molecules meets the mass conservation?
  6. Now the article has a different conclusion from the first one particularly on the effect of diameter on propulsion performance. However, the new conclusion also does not seem to be supported by data with only one case that shows 0.2% decrease from the case next to it. It could simply be a numerical error. As far as this reviewer's concerned, the data does not clearly support the conclusion.
  7. How about the effect of the length? The length is the one of main parameters of this study on the effect of combustion chamber’s structure parameters. Does the article still insist that further extension of the length will improve propulsion performance without any limit?

Author Response

Thank you for your comments on this article.Please see the attachment for the specific response comments.
